

# Role of K-feldspar and quartz in global ice nucleation by mineral dust in mixed-phase clouds

Marios Chatziparaschos[1,2], Nikos Daskalakis[3], Stelios Myriokefalitakis[4], Nikos Kalivitis[1], Athanasios Nenes[2,7], María Gonçalves Ageitos[5,6], Montserrat Costa-Surós[5], Carlos Pérez García-Pando[5,10], Medea Zanoli[3], Mihalis Vrekoussis[3,8,9] and Maria Kanakidou[1,2,3]*

[1]Environmental Chemical Processes Laboratory (EPCL), Department of Chemistry, University of Crete, Heraklion
[2]Center for the Study of Air Quality and Climate Change (C-STACC), Institute of Chemical Engineering Sciences (ICE-HT), Foundation for Research and Technology, Hellas) (FORTH), Patras, Greece
[3]Laboratory for Modelling and Observation of the Earth System (LAMOS), Institute of Environmental Physics (IUP), University of Bremen, Bremen, Germany
[4]Institute for Environmental Research and Sustainable Development, National Observatory of Athens (NOA), GR-15236 Palea Penteli, Greece
[5]Barcelona Supercomputing Center (BSC), Barcelona, Spain
[6]Department of Project and Construction Engineering, Universitat Politècnica de Catalunya – Barcelona TECH (UPC), Barcelona, Spain
[7]Laboratory of Atmospheric Processes and their Impacts (LAPI), School of Architecture, Civil and Environmental Engineering (ENAC), Ecole Polytechnique Federale de Lausanne, Switzerlandv
[8]Center of Marine Environmental Sciences (MARUM), University of Bremen, Germany
[9]Climate and Atmosphere Research Center (CARE-C), The Cyprus Institute, Nicosia, Cyprus Laboratory of Atmospheric
[10]ICREA, Catalan Institution for Research and Advanced Studies, Barcelona, Spain

*Correspondence to*: Maria Kanakidou (mariak@uoc.gr)

**Abstract.** Ice formation is enabled by Ice Nucleating Particles (INP) at higher temperatures than homogeneous formation and can profoundly affect the microphysical and radiative properties, lifetimes, and precipitation rates of clouds. Mineral dust emitted from arid regions, particularly potassium-containing feldspar (K-feldspar), has been shown to be the most efficient INP through immersion freezing in mixed-phase clouds. However, despite quartz having a significantly lower ice nuclei activity, it is more abundant than K-feldspar in atmospheric desert dust, and therefore may be a significant source of INP. In this contribution, we test this hypothesis by investigating the global and regional importance of quartz as a contributor to INP in the atmosphere relative to K-feldspar. We have extended a global 3-D chemistry transport model (TM4-ECPL) to predict INP concentrations from both K-feldspar and quartz mineral dust particles with state-of-the-art parameterizations using the ice nucleation active surface site approach for immersion freezing. Our results show that K-feldspar remains the most important contributor to INP concentrations globally, but also the contribution of quartz can be significant, for example reaching up to 60% over Eurasia at 700hPa and up to 4 across the middle and high latitudes of the Southern Hemisphere. Importantly, our calculations show that quartz may affect different cloud level regimes (low-level clouds) than K-feldspar (mid-level clouds). Additionally, the consideration of quartz INP improves the comparison between simulations and observations at low temperatures. Our simulated INP concentrations predict ~51% of the observations gathered from different campaigns within





an order of magnitude and ~69% within one and a half order of magnitude, despite the omission of other potentially important INP aerosol precursors like marine bioaerosols. All in all, our findings support the importance of considering quartz in addition to K-feldspar as INP in climate models and highlight the need of further constraining their abundance in arid soil-surfaces

along with their abundance, size distribution and mixing state in the emitted dust atmospheric particles.

## 1 Introduction

In the atmosphere, aerosol particles are emitted from various natural and anthropogenic sources or formed from precursor gases. Aerosols are important for climate due to their interactions with radiation, by scattering and absorbing light (direct climate effect), and their ability to act as precursors of clouds, modulating clouds' global distribution and radiative forcing

(indirect effect) (IPCC, 2021), along with the hydrological cycle (Seinfeld et al., 2016). The radiative properties and microphysics of clouds are particularly sensitive to aerosol concentration, composition and size (Boucher, 2015). Aerosols serve as cloud condensation nuclei (CCN) to form liquid cloud droplets (Seinfeld et al., 2016) or as ice nucleating particles (INP) to form ice crystals (Murray et al., 2012). Ice nucleation in clouds proceeds by homogeneous freezing of liquid droplets at temperatures lower than −37°C (Pruppacher et al., 1998a) or by heterogeneous freezing, triggered by INP at warmer

temperatures (DeMott et al., 2010; Murray et al., 2012; O'Sullivan et al., 2016; Price et al., 2018). INP can dramatically reduce the lifetime of shallow clouds (Vergara-Temprado et al., 2018). Also, the presence of INP may alter the development of deep convective clouds through convective invigoration (Rosenfeld et al., 2008), where enhanced glaciation of clouds increases the release of latent heat and buoyancy, which in turn increases up-draught and vertical development, thus modifying the cloud structure (Lohmann and Gasparini, 2017).


In mixed-phased clouds (MPCs), when INP concentrations are very low, the supercooled clouds are mostly composed of liquid droplets, so that CCN variations directly modulate the clouds' microphysical properties and radiative forcing. Increases in INP concentrations tend to glaciate clouds as water vapor preferentially partitions to ice owing to its lower equilibrium vapor pressure than water (Bergeron-Fiendesein process; Pruppacher et al., 1998). Given that INP tend to be orders of magnitude

lower in concentration than CCN (Seinfeld et al., 2016), a cloud that glaciates can rapidly redistribute its water from many cloud droplets down to much fewer ice crystals, leading to the formation of precipitation (Pruppacher et al., 1998b; Seinfeld et al., 2016). This entails a shorter cloud lifetime, lower albedo, less reflected radiation, and thus less cooling of the atmosphere than by liquid clouds (DeMott et al., 2010). Secondary processes (ice-ice collisions, droplet riming and droplet shattering) can modulate further these phenomena (e.g., Georgakaki et al., 2022; Sotiropoulou et al., 2021) so that ice production is promoted

even at extremely low INP concentrations (e.g., Morales Betancourt et al., 2012; Sullivan et al., 2018). High concentrations of CCN can further modulate these processes by reducing the efficiency of riming and droplet shattering (e.g., Lance, 2012). Although global aerosol models consider the aerosol-cloud droplet link (Ghan et al., 2011), much work remains to improve the description of ice formation (e.g. Vergara-Temprado et al., 2018), since the dynamics and properties of mixed-phase clouds





are among the largest uncertainties in climate models (Fan et al., 2016). Murray et al. (2021) suggest that in a warmer future climate, ice in mixed-phase clouds will be replaced by liquid water, potentially leading to strong negative feedback. However, it is uncertain whether clouds damp or amplify warming, and a key factor of this uncertainty is how models estimate INP. Many thermodynamic and microphysical feedbacks determine the glaciation state and properties of mixed-phase clouds, all of which can be affected by perturbations in CCN and INP concentrations (Solomon et al., 2018). Representing mixed-phase cloud properties in models requires an accurate description of INP along with the dynamical drivers of ice and cloud droplet formation. To achieve this, parameterizations of ice-nucleating properties of the aerosol components, which is species-specific, need to be considered in climate models (Murray et al., 2012).

While several aerosol types, like bioaerosol, black carbon and dust, have been proposed to act as INP, the most important INP in the global atmosphere is thought to be mineral dust emitted into the atmosphere from deserts and other arid and semi-arid regions (Murray et al., 2012; Seinfeld et al., 2016). This is supported by a plethora of observational and modelling studies; the most direct evidence perhaps is measurements showing that ice crystal residuals in mixed-phase clouds are strongly enhanced in mineral dust (Pratt et al., 2009). Initially, most ice nucleation studies focused on clay minerals (Hoose et al., 2010a; Roberts and Hallett, 1969); clays tend to be hydrophilic and, therefore can act as a CCN, which is a prerequisite for acting as an immersion mode INP (Kumar et al., 2011), and are transported over longer distances from the source regions due to their higher abundance in the smallest dust sizes and hence longer lifetime (Ginoux, 2017; Kok et al., 2017). More recently, it was shown that among dust minerals, K-feldspar is the most efficient INP (Murray et al., 2012; Hoose et al., 2010b; Niemand et al., 2012; Atkinson et al., 2013; Vergara-Temprado et al., 2017; Harrison et al., 2019 and references there in). To explain the enhanced K-feldspar ice activity, it has been suggested that surface -OH groups on feldspars exposed to surface defects i.e. steps, cracks, and cavities, are responsible for its high ice nucleation efficiency by allowing ice-like structures to form (Kiselev et al., 2016). However, other minerals in the airborne dust remain insufficiently characterized regarding their ice nucleating activity. Quartz is a major component of airborne mineral dust but has lower ice nucleation ability than K-feldspar, with approximately 2 orders of magnitude less active site density. Several studies have shown that quartz can also be a significant source of INP (Harrison et al., 2019; Kumar et al., 2018; Senthil Kumar and Rajkumar, 2013). Boose et al. (2016) investigated the representativeness of surface-collected dust for the airborne dust particles and showed a correlation between the INP activity of nine desert dust types and the combined concentration of quartz and K-feldspar, emphasizing the importance of taking into account the presence of both quartz and K-feldspar in the atmosphere for INP modelling.

Several approaches have been proposed to parameterize INP in climate models (e.g. Lohmann and Diehl, 2006; Spracklen and Heald, 2014; Wang et al., 2014). The first empirical parameterizations to calculate heterogeneous ice nucleation assumed that INP concentrations depend on i) temperature only (Meyers et al., 1992) or ii) temperature and the concentration of aerosol particles with a diameter larger than 0.5 μm (DeMott et al., 2010). These parameterizations were the first attempts to estimate INP concentrations and did not consider additional important complexities such as chemical composition variations over time



and space. The simplest and most performant parameterization is based on ice active sites density concept (Vali et al., 2015). This approach adopts the so-called singular hypothesis, where the time dependence (stochasticity) of nucleation is assumed to

be of secondary importance, which should be a good approximation if the site density is large enough (Vali et al., 2015). This hypothesis allows formulating a very simple link between aerosol chemical composition (surface structure), size distribution and an ice nucleation spectrum constrained by experimental data (e.g., Barahona and Nenes, 2009), from which one can predict INP as a function of supersaturation (in the deposition mode) or temperature (in the immersion mode). Immersion freezing is thought to be the most dominant freezing mechanism (Westbrook and Illingworth, 2013) in mixed-phase clouds. This is the

pathway we focus on in this study. Different materials are assumed to have a different ice active density ($n_s$) per unit surface and a critical temperature below which an INP with multiple nucleation sites activates in the immersion mode to ice (Hoose et al., 2010a; Murray et al., 2012). Only a few INP parameterizations in immersion freezing mode explicitly consider $n_s$, and only recently, models that adopt this level of compositional complexity have emerged, mainly considering one type of INP from the most active mineral K-feldspar (Atkinson et al., 2013; Vergara-Temprado et al., 2017).


Harrison et al. (2019) proposed a new parametrization based on experimental work in immersion freezing mode by testing 10 naturally occurring quartz samples that highlighted the variability in their ice-nucleating ability. It was observed that the quartz group of minerals is generally less active than K-feldspars, with the most active quartz samples being of similar activity to some K-feldspars. It was also found that the ice-nucleating activity of some quartz samples is very sensitive to ageing when

exposed to air and water, contrary to other quartz and K-feldspars samples. This variability indicates that mineral dust particles do not nucleate ice through a lattice-matching mechanism. The latter is consistent with recent observations that ice nucleation occurs at active sites (Holden et al., 2019). However, once in the atmosphere, mineral dust particles undergo ageing through the condensation of sulfates, nitrates and secondary organic aerosol material, becoming soluble and thus promoting wet scavenging. The occurrence of such atmospheric ageing of INP via coating by acids or water-soluble organics that modifies

the ice-nucleating efficiency of aerosols is supported by the detection of sulphate and/or nitrate peaks of the IN active particles using single particle analysis by micro-Raman spectroscopy (Iwata and Matsuki, 2018) but the study of the impact of this process on INP is beyond the scope of the present study.

Here we present, to our knowledge, the first global modelling studying the role of airborne mineral dust as a precursor of INP,

accounting for both its quartz and feldspar contents. We simulate the three-dimensional global distribution of INP concentrations, evaluate the model results against observations, and investigate a) the importance of quartz relative to feldspar in INP formation and b) potentially missing INP sources other than dust minerals. To achieve this, we use a well-documented global 3-D chemistry-transport model to simulate the global temporally and spatially varying immersion-mode distributions of INP from feldspar and quartz minerals.



## 2 Methods

### 2.1 Global modelling

For the present study, we use the global 3-dimensional chemistry-transport model TM4-ECPL (Daskalakis et al., 2016, 2022; Myriokefalitakis et al., 2016, 2015; Kanakidou et al., 2020) driven by ERA-interim reanalysis meteorological fields produced with the European Centre for Medium-Range Weather Forecasts (ECMWF) meteorological model (Dee et al., 2011). The specific model version has a horizontal resolution of 3º longitude by 2º latitude with 25 hybrid pressure vertical levels from surface up to 0.1hPa (about 65 km) and uses a model time step of 30 minutes. The model simulates tropospheric chemistry accounting for non-methane volatile organics, all major aerosol components, including dust, and the atmospheric cycles of nitrogen, iron and phosphorus (Tsigaridis et al., 2014; Tsigaridis and Kanakidou, 2007; Kanakidou et al., 2016, 2020; Myriokefalitakis et al., 2015, 2016). Uncertainties in the simulations resulting from the use of different emission databases have been investigated by Daskalakis et al. (2015). The model considers lognormal aerosol distributions in fine and coarse modes and allows hygroscopic growth of particles, as well as removal by large-scale and convective precipitation and gravitational settling. In-cloud and below-cloud scavenging are parameterized in TM4-ECPL as described in detail by Jeuken et al. (2001). In-cloud scavenging of water-soluble gases accounts for the solubility of the gases (effective Henry law coefficients; Myriokefalitakis et al., 2011; Tsigaridis et al., 2006 and references therein). Details on the emissions used in the model are provided in Myriokefalitakis et al. (2015, 2016) and Kanakidou et al. (2020). Dust emissions are calculated online as described by Tegen et al. (2002) and implemented as in Van Noije et al. (2014). Dust aerosol (including feldspar and quartz) emissions are represented by lognormal distributions in fine and coarse modes with dry mass median radii (lognormal standard deviation) of 0.34µm (1.59) and 1.75µm (2.00), respectively. We assume that all aerosols are externally mixed. Typical lognormal distribution equations are used to convert mass to number concentration of aerosols in each grid box of the model.

### 2.2 K-feldspar and quartz emissions

To calculate the quartz and K-feldspar emissions we use the global soil mineralogy atlas of Claquin et al. (1999), including the updates proposed in Nickovic et al. (2012). The atlas of Claquin et al. (1999) provides mineral fractions for eight major minerals, including quartz and feldspar, in arid and semi-arid regions for two soil size fractions: clay (up to 2µm) and silt (from 2 to 50µm). Claquin et al. (1999) exclusively apportion feldspar to the silt size fraction, but feldspar is also observed in the smaller sizes, both in the soil (e.g. Journet et al., 2014) and the airborne dust (e.g. Kandler et al., 2009). Following Atkinson et al. (2013) we added feldspar in the clay size fraction of the soil by scaling the feldspar fraction in the silt size fraction of the soil with the clay-to-silt ratio of quartz. Since Claquin et al. (1999) only estimated total feldspar, based on observations, we also assumed that K-feldspar represents 35% of feldspar (Atkinson et al., 2013).

Atkinson et al. (2013) calculated the emission of K-feldspar as the product of dust emission flux and the soil fractions of K-feldspar. Here, we additionally account for the known differences between soil and emitted mineral fractions. The soil samples that constitute the basis of soil mineralogical atlases are subject to destructive analytical techniques, e.g., wet sieving. As a



result, the aggregates originally present in the soil are more effectively disturbed than through wind erosion processes, hence overemphasizing the fine fraction of clay minerals compared to their actual abundance at emission. We calculated the emitted mass fractions of quartz and K-feldspar in the fine and coarse modes based on Brittle Fragmentation Theory (BFT) from Kok (2011). BFT restores soil clay-sized minerals (mainly illite, kaolinite and smectite), created by wet sieving, into emitted coarse aggregates. As a consequence, the fractional contribution of quartz and K-feldspar in the emitted coarse dust is reduced relative to their fractions in the silt size fraction of the soil (Pérez García-Pando et al., 2016; Perlwitz et al., 2015a, b). Finally, the corresponding accumulation and coarse-mode emission of each mineral is calculated by applying the respective mineral emitted mass fractions to the dust emission flux.

## 2.3 Calculation of INP concentrations

To consider the influence of mineralogy on INP concentrations in TM4-ECPL, the singular approximation of ice-nucleating activities of aerosols is used, based on laboratory-derived active site parameterizations provided in Harrison et al. (2019). The parameterization for K-feldspar captures the observed plateau in active sites density ($n_s(T)$) below −30 ºC, reproducing higher $n_s(T)$ values at temperatures warmer than −10˚C, additionally, Harrison et al. (2019) provide a new parameterization for quartz minerals to constrain their role as ice nuclei particles. The parameterisations are valid for a temperature range from −3.5 to −37.5ºC for K-feldspar and -10.5 to -37.5ºC for quartz, and they have an uncertainty, expressed as the standard deviation for log($n_s$(T)), of about 0.8 for both quartz and K-feldspar INPs that corresponds to an order of magnitude in their concentrations. Part of this uncertainty is derived from impurities that influence how individual quartz samples fracture, impacting the presence of active sites. In this study, the parameterisations are exclusively applied within the valid temperature limits since outside these limits, the parameterizations yield non-realistic values that we have excluded from our estimates. Therefore, in this study INPs from quartz and from K-feldspar are not calculated beyond the respective temperature limits.

We note that the ice-nucleating ability of the different minerals is determined from laboratory experiments using artificial soil samples (Kumar et al., 2018), which may not be representative of realistic atmospheric conditions. For instance, they omit important atmospheric processes such as ageing from acids and regeneration of small airborne dust particles, which most likely have reformed surfaces. Data analysis provided in Atkinson et al. (2013) suggests that aggregation of feldspar particles resulting in reduced surface area is at most a minor effect compared with the differences in active sites density of different types of K- feldspar (factor of 6) (Harrison et al., 2016). Additionally, considerable differences in simulated INP numbers using laboratory-based and field-based INP parameterisations are attributed to biological nanoscale fragments attached to mineral dust particles (O'Sullivan et al., 2016). Such effects of atmospheric ageing on INP distributions are not considered in the present study.

The calculation of INP number concentrations follows the ice nucleation active surface site density concept of Vali et al. (2015) and assumes that the freezing of supercooled droplets is, to a first approximation, time-independent and that the ice germ formation is induced by specific sites on the aerosol surface (singular description). The ice nucleation active surface site density


is derived by the spectrum of ice-nucleating properties and for a polydisperse aerosol sample is given by the Poisson
distribution:

$$\sum_{j=1}^{k} n_{i,j}(T, S_{i,j}) = \sum_{j=1}^{k} n_{aer,j}\{1 - exp[-S_{aer,j}\, n_s(T, S_{i,j})]\} \tag{1}$$

Where $n_{i,j}$ is the ice number concentration formed on the aerosol, indexes i, j correspond to aerosol type (quartz or K-feldspar)
and size mode (accumulation or coarse mode), respectively. $S_{i,j}$ is the individual aerosol particle mean surface area in size
mode j and $n_{aer,j}$ is the total aerosol number concentration (see Niemand et al. (2012) for details). In the present study, we use
the $n_s$ parameterizations provided in Harrison et al. (2019) that are good approximations for the majority of feldspars and
quartz studied in the laboratory.

In order to estimate aerosol surface area, we assume that each mineral dust particle is spherical and externally mixed (Atkinson
et al., 2013; Vergara-Temprado et al., 2017). As discussed in Harrison et al. (2019), Atkinson et al. (2013) found that fully
internally mixed feldspar particles produce the same INP concentrations with partially internally mixed particles at
temperatures above -25°C but lead to higher INP concentrations by one to two orders of magnitude at the lowest temperatures
of validity of the parameterization. In addition, in the present study, removal mechanisms by physical processes are applied
directly to all dust particles such as sedimentation, deposition and scavenging and not preferentially to INPs.

There are two different ways to represent simulated INP concentrations by the model, either as potential INP that can activate
to ice at a fixed temperature ([INP]$_T$) or as ambient INP that are calculated to be active in the model's ambient local atmospheric
temperature. [INP]$_{ambient}$ is a useful way of looking at the INP concentration relevant to non-deep convective mixed-phase
clouds, while [INP]$_T$ can be an indicator for the distribution of INPs concentrations at temperatures that will affect clouds with
vertical extent like deep-convective systems (Vergara-Temprado et al., 2017) as illustrated in Figure 1. Additionally, because
most measurements are made by exposing particles to specific temperatures depending on instrumentation, [INP]$_T$ is a pertinent
quantity when comparing modelled and observed INP concentrations.

## 3. Evaluation of dust and INP simulations

### 3.1 Dust

Airborne dust concentration is of paramount importance to this work, since modelled dust affects directly the amount of INP
in the atmosphere. Consequently, we evaluate modelled dust by comparing with monthly averaged dust observations from
stations that are in the outflow of Saharan desert and with worldwide climatological observations. The climatological annual
means comparison allows us to validate the geographical distribution of sources and transported dust regions, while the
monthly means comparison reveals if the intra-annual variability can be well captured by the model.





We first evaluate the dust mass simulations against monthly averaged dust observations at nine ground-based stations between 2000 and 2017 (Figure S4). All stations are in areas suitable for studying emissions of Saharan dust and its atmospheric transport. Four stations (M'Bour, Bambey, Cinzana, and Banizoumbou) are located at the edge of the Sahel, the major natural dust source of the region (Lebel et al., 2010), three stations (Miami, Barbados and Cayenne) are on the American continent downwind the dust Atlantic transport (Zuidema et al., 2019; Prospero et al., 2020) and two stations, Finokalia (Crete, Greece) (Mihalopoulos et al., 1997; Kalivitis et al., 2007) and Agia Marina (Cyprus) (Pikridas et al., 2018; Kleanthous et al., 2014), lie on the dust transport route crossing the Mediterranean. Figure 2 depicts the comparison of model dust aerosol concentrations with observations on a monthly mean basis (correlation coefficient, R=0.81). The methodology to derive dust concentrations from the aerosol observations at these sites is discussed in the supplement. The mean bias between model and observations is 8.3 $\mu g \cdot m^{-3}$. The total average dust concentration observed at the selected locations and during the studied period 2000-2017 is approximately 43.3 $\mu g \cdot m^{-3}$. Thus, the mean bias corresponds to 5% of the total concentration. The relative error increases with decreasing concentrations, and only a few outliers are not within one order of magnitude of the observations in Figure 2, mainly corresponding to observations at the Caribbean and American stations.

We further evaluate modelled dust by comparing with climatological globally distributed observations of dust surface concentration and deposition for the years from 2009 to 2016 when we also perform the comparison between observed and modelled INP concentrations (see below, 3.2). Climatological annual means of dust surface from the Rosenstiel School of Marine and Atmospheric Science (RSMAS) of the University of Miami (Arimoto et al., 1995; Prospero, 1996, 1999) and the African Aerosol Multidisciplinary Analysis (AMMA) (Marticorena et al., 2010) are compared with the multi-annual model mean surface dust concentration (Figure 3a). We also compare the modelled deposition dust fluxes with the observations for modern climate compiled by Albani et al. (2014). Measurements from 110 different locations and the constrained mass fraction for particles with a diameter lower than 10μm to conform to the range of simulated sizes are compared to modeled deposition dust fluxes (Figure 3b). Table S1 summarizes the statistical parameters that are used to validate the model and the locations (and regions) of the various observations are depicted in Figure S1. The correlation coefficient (*R*) reveals the linear relationship between model results and observations, while the normalized mean bias (nMB), and the normalized root-mean-square error (nRMSE) are the measure of the mean deviation of the model from the observations due to random and systematic errors. All the equations that are used for the statistical analysis of model results are provided in the Supplement (Eqs. S1–S3).

The geographical distribution of the available concentration stations (23 in total) in Figure 3a covers locations close to emission sources (AMMA stations over the Sahelian dust transect), and in both transport and remote regions (RSMAS). The overestimation of the dust surface concentration already shown at the monthly scale is also observed in the comparison with annual mean climatological observations, with an overall normalized mean bias of 44.1%. Errors are generally larger after transport to remote regions, particularly over the North Pacific, than over source regions (Figure 3a). When it comes to deposition, the errors are larger than in the representation of the surface concentration fields, particularly overestimations are


found over the Southern Ocean and Arctic, than downwind sources (Figure 3b); while Europe and South America are largely underestimated. They are, however, kept within the one-order of magnitude differences, except for those points located over

the Southern Ocean, where overestimations are significant. The aerosol concentrations and deposition fluxes in such remote regions are usually low and thus, small differences are emphasized in relative terms. There may be multiple causes for the discrepancy between modelled and observed fluxes (e.g. an excessive modelled fine fraction of aerosols, differences in the wet deposition fluxes caused by differences in precipitation, differences in the observed and modelled period, or uncertainties associated to the observations themselves). The main purpose of this comparison is to assess the ability of TM4 to reproduce

the geographical distribution of dust in the atmosphere, which is found satisfactory, and exploring further the issues behind the deviation in a particular remote region is out of the scope of this work.

## 3.2 INP

In Figure 4, we compare the INP concentrations calculated by the TM4-ECPL model with available INP observations from

the BACCHUS (Impact of Biogenic versus Anthropogenic emissions on Clouds and Climate: towards a Holistic UnderStanding) (http://www.bacchus-env.eu/in/index.php, last access on March 2019) and Wex et al. (2019) (https://doi.pangaea.de/10.1594/PANGAEA.899701, last access on February 2022) databases. These data span the years 2009 to 2016 and originate from different campaigns (Figure S3, supplement). For this comparison, $[INP]_T$ concentrations are calculated at the temperature at which measurements were performed. All observations are compared with the same temporal

resolution (same month and year) of the model, except the observed dataset by Yin et al.(2012), which covers temporally scattered measurements (between 1963 to 2003). This dataset is compared to the modeled multi-annual monthly mean INP concentration considering the years 2009 to 2016. The color bar shows the corresponding temperature at which the measurement is performed (Figure 4a) or the modeled dust aerosol mass concentration (Figure 4b). These different color bars allow us to determine if the bias of the model can be correlated with the temperature of the observed INP (e.g., highly active

INP reddish symbols in Figure 4a) or with modelled dust mass (e.g., low modelled dust mass, blueish symbols in Figure 4b). The majority of simulated $[INP]_T$ are within 1.5 orders of magnitude of the observed values as depicted in Table S2.

The model is not efficient enough in simulating highly ice-active INP as presented in Figure 4a (red spots corresponding to high temperatures) observed in both Wex (triangles) and BACCHUS (circles) databases. The temperature range of these

measurements is very close to the temperature limits of the Harrison et al. (2019) parameterization. Since the quartz parameterization temperature limit is at -10.5°C, INP observations measured between -3.5 and -10.5°C are compared with simulated INP derived from K-feldspar only. Supplementary Figure S7 depicts the comparison to observations using quartz and K-feldspar separately. This figure reveals that INP from quartz can improve the comparison with observations at low temperatures, which is consistent with Boose et al. (2016) that suggested that for temperatures between -33°C to -37°C quartz





can be a significant contributor to INP concentration. Figure S7d clearly shows that consideration of quartz together with Feldspar INP improves the comparison with observations by moving the respective points closer to the 1:1 line. This is particularly true for high INP concentrations at low temperatures and for relatively low INP concentrations at temperatures around -20°C. Overall about 3% more data points are within 1.5 orders of magnitude from the observations when accounting quartz INP in addition to feldspar INP (Table S2). Additionally, modeled INP concentrations are strongly underestimated

compared to observations for temperatures above -25°C. Si et al. (2019) pointed out that mineral dust was a major contributor to the INP population at temperatures lower than -25°C. They also found that, for three coastal sites, modeled INP concentrations based on K-feldspar as the only INP precursor agreed well with INP measurements at -25°C, but measurements at −15°C were underestimated, indicating a missing source of INP that activates at temperatures higher than -25°C. These results on the importance of dust as INP for different temperature ranges agree with our study as can also be seen in

Supplementary Figure S7.

Figure 4b reveals that the observed high concentrations of INP correspond to high dust mass concentrations. Additionally, the observed low concentrations of INP correspond to particles that activate at relatively high temperatures (higher than -10°C; reddish symbols in Figure 4a) and low mass dust concentration (blueish symbols in Figure 4b). Thus, our model underestimates

measurements corresponding to low dust mass concentrations (blueish symbols in Figure 4b). As shown in Figure 4a, these points correspond to measurement temperatures around -20°C and -25°C. Therefore, these INP observations not reproduced by our model when accounting only for dust-originating INP are probably not solely affected by airborne mineral dust, pointing to other aerosol sources contributing to INP and especially marine bioaerosols. Indeed, mineral dust particles likely only become ice active at low temperatures, but they may be carriers of biogenic ice-active macromolecules (O'Sullivan et al.,

2014, 2016; Hill et al., 2016), which enhance dust nucleation activity at higher temperatures. Ice nucleation at temperatures in the vicinity of zero is typically related to macromolecules from biogenic entities such as bacteria, fungal spores, pollen, and marine biota. These ice-active macromolecules nucleate ice from just below 0°C down to roughly −20°C (Murray et al., 2012; Kanji et al., 2017; O'Sullivan et al., 2018). Indeed, based on air mass back-trajectories analysis, Wex et al. (2019) (triangles in Fig 4a) suggested that both terrestrial locations in the Arctic and the adjacent sea were possible source areas for highly active

INPs derived from material of biogenic origin.

Our model agrees reasonably well with observations of INP (R=0.84, Table S2) at the full temperature range of the measurements when dust is present in significant amounts (dust concentrations larger than $10^{-1}$ μg m$^{-3}$) (Figure 4b) and at temperatures lower than -10°C (Figure 4a). Overall, the model reproduces about 51% of the observations within an order of

magnitude and about 69% within one and a half orders of magnitude (Table S2). Supplementary Figure S2 depicts globally the regions where the model overestimates (blueish) and underestimates (reddish) INP observations. Improving the representation of the dust cycle, as discussed in section 3.1.1 (Figure 3) could reduce the bias of INP simulations. The model underestimates the dust surface concentrations observations over Europe and Southern Ocean (Figure 3a) as well as the dust





deposition fluxes over South Africa and Western Asia (Figure 3b), respectively as shown in Table S1. This may at least partly
explain the underestimation by the model of the observed INP concentrations (close to 2 orders of magnitude, reddish points)
in these regions, as depicted in Supplementary Figure S2. Additionally, the boxes in Supplementary Figure S2 depict the
climatological monthly mean bias (observations – model) at specific regions (Arctic, Europe, China, North America, Central
America, North Africa, South Africa and West Asia-Middle East). This analysis shows that observed INP concentrations are
underestimated by 2 orders of magnitude over Europe during autumn and over West Asia-Middle East during autumn and
winter.

The overall agreement between the model results and observations is reasonable (see also Supplementary Figure S8), but there
are significant discrepancies. These differences between model results and observations could be attributed to several factors,
such as the omission of the contribution of marine organic (Spracklen and Heald, 2014; Wilson et al., 2015) and terrestrial
biogenic aerosols (Chatziparaschos et al., 2021). Although some biological nanoscale fragments could be attached to mineral
dust particles (Froehlich-Nowoisky et al., 2015; O'Sullivan et al., 2016; Violaki et al., 2021), their effect on the ice nucleating
activity of dust was not considered here. Additionally, model deviation from measurements could be attributed to potential
inaccuracies in the simulated content of quartz and K-feldspar due to uncertainties in the dust cycle and in the emitted mineral
fractions (Perlwitz et al., 2015; Pérez García-Pando et al., 2016). Furthermore, errors may be partly due to atmospheric
processes such as ageing, resulting in ice ability degradation (leading to positive model bias) or a possible underestimation of
the concentration of these transported aerosol species (leading to negative model bias). These processes are currently neglected
in our model but the sensitivity of the INP simulations to these uncertainties is the subject of an ongoing complementary study.

## 4. K-feldspar and quartz contributions to the global INP distribution

Figure 5a presents the distributions of particles at a pressure of 800hPa able to freeze in the immersion mode at -20°C, hereafter
called potential INP, $[INP]_{-20}$, as derived from K-feldspar and quartz simulated concentrations. These conditions could be
representative of mixed-phase clouds' glaciation and allow both quartz and K-feldspar to activate and form INP. As stated
earlier, the $[INP]_{-20}$ distributions are of interest for their impact on clouds formed in deep-convective systems that are
experiencing large vertical velocities (Figure 1). Such systems are observed in subtropical and tropical regions. In addition,
when comparing measurements of [INP] concentration to simulated [INP], we compare these quantities at specific activation
temperatures. Figure 5a shows that $[INP]_{-20}$ concentrations derived from K-feldspar and quartz contribute much more to mid-
latitude, mid-level mixed-phased clouds in the Northern Hemisphere (NH) than in the Southern Hemisphere (SH) due to dust
source location and long-range atmospheric transport patterns. However, there are also dust-originating INP sources in the SH,
such as those from Patagonia, South Africa and west Australia, that yield considerable $[INP]_{-20}$ concentrations (larger than $10^{-2} L^{-1}$).






To investigate the spatial variability of the contribution of each mineral to potential INP concentration, we also present the ratio of INP from quartz ([INP]$_{quartz}$) to the INP derived from both quartz and K-feldspar ([INP]$_{total}$) at -20ºC (Figure 5b). This analysis shows regions where [INP]$_{quartz}$ has the potential, under these meteorological conditions, to contribute over 30% of the total [INP]$_{-20}$ concentration over North Atlantic, south USA and more than 40% over India and Eurasia.


More specifically, the North Atlantic and generally the NH are more likely affected by ice formation from both quartz and K-feldspar. In contrast, the South Atlantic is affected by K-feldspar INP originating from the Patagonia desert since 75% of [INP]$_{-20}$ are derived from K-feldspar, and the remaining 25% is from quartz (Figure 5b). Additionally, there are quartz INP sources in South Africa from the Kalahari Desert, since 35% of the total INP concentrations (around $10^{-2}$L$^{-1}$) are originated from quartz

and high percentages of quartz minerals and low feldspar (about 6%) in terms of mass are calculated, as is depicted in Figures 5c-d.

The areas most affected by INP derived from quartz are the regions of the South Saudi Arabian Peninsula, India, North Indonesia and Eurasia (Figure 5b), where high ratios of quartz minerals to dust mass (Figure 5d) and high concentrations of

INP of about 0.5 L$^{-1}$ are simulated (Figure 5a). To these dust particles that are probably originating from Saudi Arabia and the Gobi Deserts, the mass contribution of quartz exceeds 35% (Figure 5d), and its contribution to INP is 35-40% (Figure 5b). The mass ratio of these minerals, calculated as the ratio of the mass of quartz to the mass of feldspar, is depicted in Figure 6. The combination of Figures 5b and 6 shows that when the mass of quartz is six times that of feldspar, the quartz INP concentration ([INP]$_{quartz}$) constitutes more than 35-45% of total INP concertation.


As expected, [INP]$_{ambient}$ ([INP] at local model temperature) concentrations follow the air temperature distribution, showing the highest modeled concentrations roughly poleward of 50 degrees in both hemispheres (Figure 7 for model temperature at 600hPa). The contributions of both minerals to INP are comparable (Figure 7c-d); [INP]$_{quartz}$ appears as important as [INP]$_{feldspar}$ around 40º and poleward of 70º. Overall, the number concentration of total [INP]$_{ambient}$, i.e. of primary ice formed by immersion

freezing, strongly depends on the ambient temperature and the mass concentrations of these minerals, with positive dependence on mass and negative dependence on temperature.

Figure 8a shows the annual zonal mean distribution of INP derived from both minerals. Total INP ([INP]$_{total}$) maximizes approximately at 600-500hPa (Figure 8a), where K-feldspar derived INP is the largest fraction of INP concentration (Figure

8b). This is attributed to the high ice-activity of K-feldspar and its temperature dependence. However, low [INP]$_{total}$ concentrations (<$10^{-2}$L$^{-1}$), which are mainly derived from quartz dust particles (60%), as shown in Figure 8b, are calculated at lower altitudes (Figure 8a). This is attributed to quartz's high number concentration partially associated with local [INP]$_{quartz}$ sources, such as the Gobi Desert (Figure 5d) in between 30-40ºN at 700 hPa and lower. INP number concentration depends on the particle number concentration of these minerals and ambient temperature. At temperatures below −25°C, the quartz





contribution becomes increasingly important with decreases in temperature at high INP concentrations. This contribution is up to 50% at −35°C (Figure 8a). These findings agree well with Ilić et al.(2022) and Boose et al. (2016) who showed that at temperatures between the homogeneous freezing limit and −33°C, quartz could be a significant contributor to [INP]. Additionally, the contribution of quartz at low INP concentrations could be explained by the high number concentration of quartz, since it is more abundant than K-feldspar, compensating for the lower ice nucleating activity of quartz with respect to

K-feldspar. Overall, $[INP]_{quartz}$ dominates at lower altitudes than K-feldspar (Figure 8b; reddish colors) and is far from emission sources. This is clearly demonstrated in Supplementary Figure S5 for Eurasia. Consequently, $[INP]_{feldspar}$ is expected to affect mid-altitude clouds such as altostratus, and altocumulus, while $[INP]_{quartz}$ is expected to affect high clouds such as cirrus clouds and low-altitude clouds. The present study emphasizes that INP concentrations could be significantly affected not only by K-feldspar as generally thought but also by quartz. The evaluation of the climate impact of these INPs is a subject of ongoing

research, and it requires the incorporation of parameterizations such as those used here into a climate model able to simulate ice formation and its interactions within the climate system.

## 5. Conclusions

The contribution of K-feldspar and quartz dust minerals as ice-nucleating particles has been investigated in this study using a three-dimensional global chemistry transport model as a first step towards their incorporation in a climate model. Laboratory-

derived parameterizations based on the singular description and aerosol chemical composition have been incorporated into the model together with an explicit representation of the dust mineralogy that considers regional variations in the composition of the dust sources and relies on the Brittle Fragmentation Theory to derive the size-distributed mineral abundances in the atmosphere. To our knowledge, this is the first time a global chemistry transport model considers the contribution of both quartz and k-feldspar to predict INP concentrations.


Quartz particles have lower ice active density than the most active mineral K-feldspar. However, quartz particles are more abundant in airborne dust than K-feldspar, partly compensating for their lower ice nucleating activity with respect to K-feldspar. We find that $[INP]_{quartz}$ dominates at lower altitudes over locations with abundant emissions but low INP concentrations. There are regions of the global atmosphere, in particular over mid-and high- latitudes of Eurasia, where quartz

contributes over 40% (at a temperature of -20ºC,800hPa) and over 60% (at ambient model temperature, 700hPa) to the total INP concentration derived from dust aerosols represented by quartz and K-feldspar. The quartz contribution becomes increasingly important with decreasing temperature, giving high INP concentrations at temperatures below −35°C and high altitudes. Consideration of quartz-derived INP is improving the comparison with observations for high INP concentrations at low temperatures and relatively low INP concentrations at temperatures around -20ºC. These results highlight the important

role of quartz in increasing INP concentrations, mainly over the mid-to-high latitudes of both hemispheres.





Overall, the simulated concentrations of INP from two-dust components agree with the observational data from the BACCHUS and Wex et al. (2019) databases within one and a half orders of magnitude. Statistical analysis of the comparison of the modeled annual mean dust surface concentration has shown that the model underestimates dust concentration over Europe and the

Southern Ocean, potentially explaining part of the bias between modeled and observed INP concentrations. Furthermore, the omission of marine and terrestrial biological INP sources, dust mineralogy uncertainties and atmospheric processes such as ageing and in-cloud removal could also explain the model's discrepancies. The evaluation of the importance of these factors on INP simulated number concentrations is the topic of ongoing studies. As importantly, the heterogeneously formed ice crystals could trigger secondary ice production processes such as the Hallett – Mossop process, which is not considered in our

study but is known to further increase the number concentration of ice crystals (Sotiropoulou et al., 2020) and their subsequent climate impact.

We conclude that dust particles appear to be globally important precursors of INP, able to account for most of the observed INP number concentration globally. K-feldspar contributes globally most of the INP associated with desert dust because it is

more active and less sensitive to ageing processes that reduce the ice-nucleating activity of the particles (Harrison et al., 2019). However, we highlight that INP concentrations could be significantly affected by quartz particles, which are abundant in airborne dust. Finally, we propose that additional experimental studies on the ice-nucleating ability of mineral dust particles must be performed to investigate the impact of atmospheric ageing and the presence of biological fragments attached to dust on the ice-nucleating activity of dust particles.


*Acknowledgements.* This work has been supported by the project "PANhellenic infrastructure for Atmospheric Composition and climatE change" (PANACEA), (MIS 5021516), which is implemented under the Action "Reinforcement of the Research and Innovation Infrastructure", funded by the Operational Programme "Competitiveness, Entrepreneurship and Innovation" (NSRF 2014-2020) and co-financed by Greece and the European Union (European Regional Development Fund). ND, MK

and MV acknowledge support from the Deutsche Forschungsgemeinschaft (DFG, German Research Foundation) under Germany´s Excellence Strategy (University Allowance, EXC 2077, University of Bremen). MK, MC, AN, MC-S, MGA and CPG-P acknowledge support by the European Union Horizon 2020 project FORCeS under grant agreement No 821205. CPG-P, MC-S, MGA acknowledge support from the European Research Council under the European Union's Horizon 2020 research and innovation programme (grant n. 773051; FRAGMENT) and from the AXA Research Fund. MC-S has received funding

from the European Union's Horizon 2020 research and innovation programme, under the Marie Skłodowska-Curie grant agreements, reference 754433 from the call H2020-MSCA-COFUND-2016. We thank J. Prospero and M. Pikridas for PM10 measurements in Banizoumbou (Niger), Cinzana (Mali), M'Bour (Senegal) were performed in the framework of the French National Observatory Service INDAAF (International Network to study Deposition and Atmospheric composition in Africa; https://indaaf.obs-mip.fr/) piloted by the LISA and LAERO and supported by the INSU/CNRS, the IRD and the Observatoire

Midi-Pyrénées and the Observatoire des Sciences de l'Univers EFLUVE.





**Data availability**

The INP data used in this study can be accessed through BACCHUS (http://www.bacchus-env.eu/in/info.php? id=71) and (frozen fractions and INP concentrations) on Pangaea (https://doi.pangaea.de/10.1594/ PANGAEA.899701; Wex et al., 2019). Model output will be available on zenodo before publication of the paper.


**Authors' contributions**

MK and MC conceived the study; MC modified the model to account for INP and performed the simulations, visualized and analyzed data, and wrote the initial version of the manuscript; MK supervised the work with feedback from AN and CPG-P and re-edited the manuscript; SM implemented the dust emissions module and the initial dataset of dust mineralogy in the

model; ND, MZ, MV and MGA evaluated the dust simulations; NK provided characterized dust aerosol observations from Finokalia station; MC-S provided key feedback on the implementation of the ice nucleation active surface site density parameterization; CPG-P and MGA provided the emitted mineral fractions of quartz and feldspar; CPG-P re-edited parts of the manuscript;  all authors read and commented the manuscript.

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




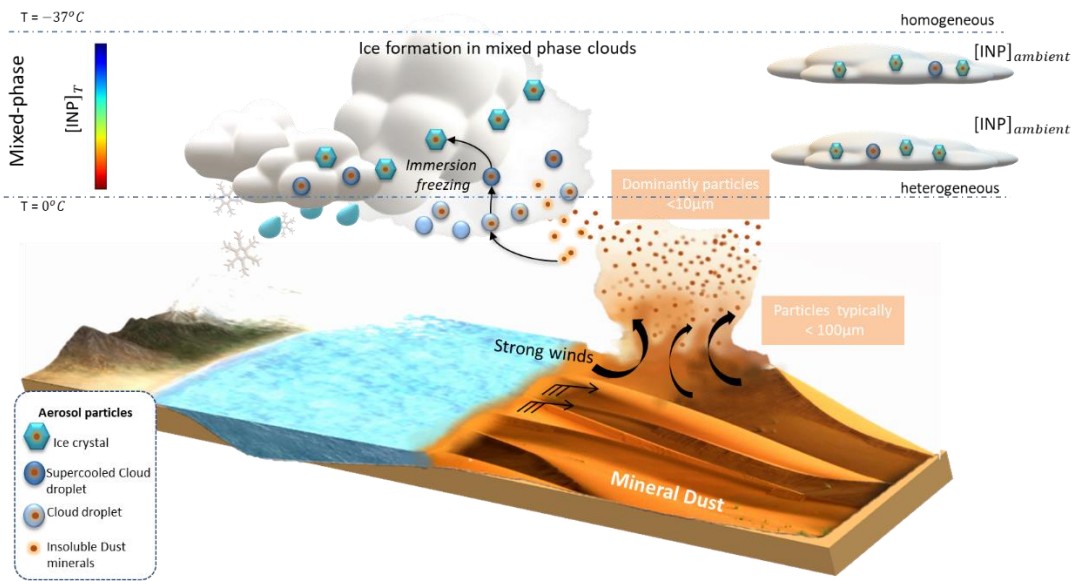

**Figure 1: Illustration of the formation of INP from mineral dust aerosol and the two ways in which we display INP concentrations in the present study: [INP]ambient calculated at ambient model temperature relevant to non-deep convective mixed-phase clouds using the ambient temperature in the model temperature level (right part of the figure), and [INP]T calculated at a fixed temperature relevant for vertically extended clouds as deep-convective systems (left part of the figure). The temperature color scale from 0 to -37oC is provided to the left (see text). Figure modified by combining the ones in Kanji et al. 2017 (Figure 1), Harrison et al. 2019 (Figure 1) and Vergara-Temprado et al. 2017 (Figure 3).**


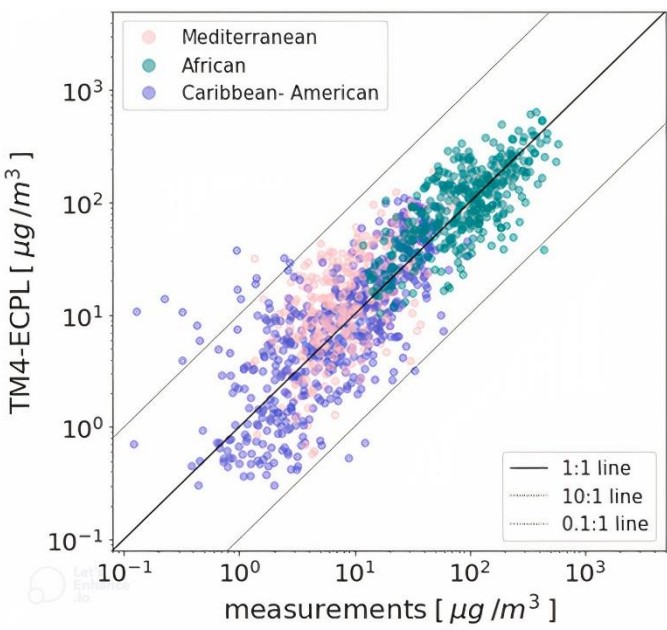

**Figure 2: Monthly averaged model-simulated versus observed dust concentration at the selected locations in µg·m⁻³. The continuous black line is the 1:1 line, while the dashed lines are the 10:1 and 1:10, top and bottom lines, respectively. The stations are geographically grouped: African stations in green (M'bour, Bambey, Cinzana, Banizoumbou), American stations in violet (Miami, Barbados, Cayenne), and Mediterranean stations in pink (Agia Marina and Finokalia). Simulations were run with a 3°x2° spatial resolution.**

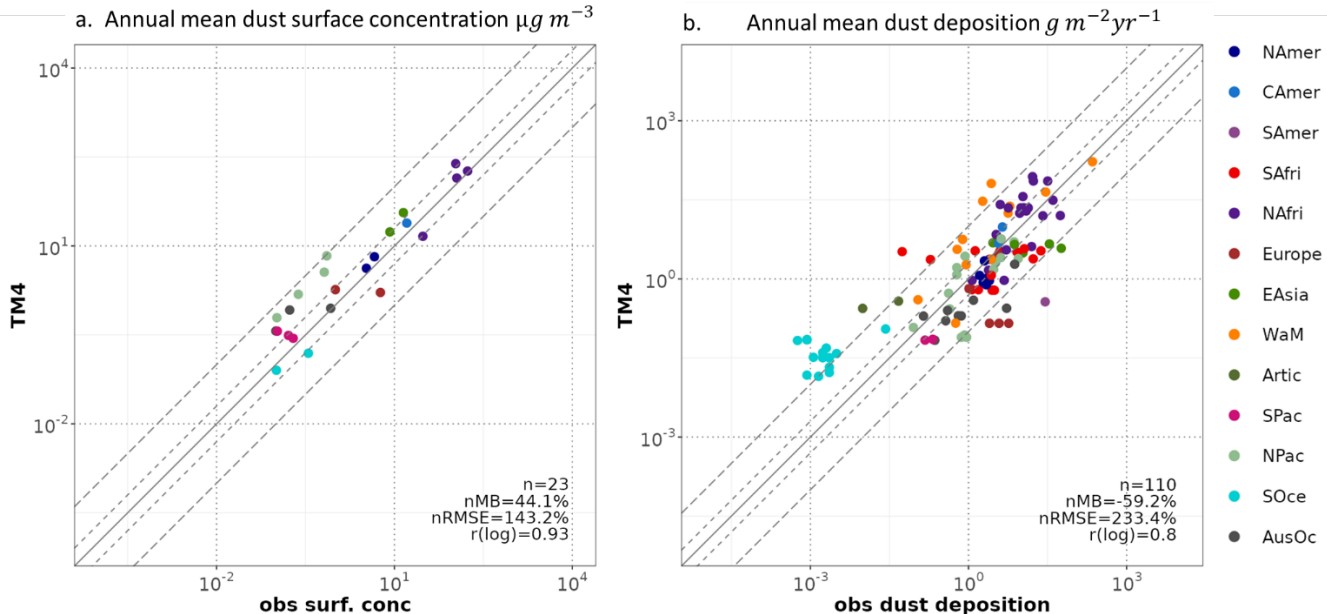

**Figure 3: Comparison of (a) the modelled annual mean dust surface concentration for the years 2009–2016 compared to**





climatological mean values from RSMAS sites and AMMA campaign, and (b) comparison of modelled annual dust deposition flux averaged for the same period against observations compiled in Albani et al. (2014) from several sources. The solid line represents
the 1 : 1 correspondence, the long dashed lines show the 10 : 1 and 1 : 10 relationships and short dashed lines corresponds to a factor of 2, respectively

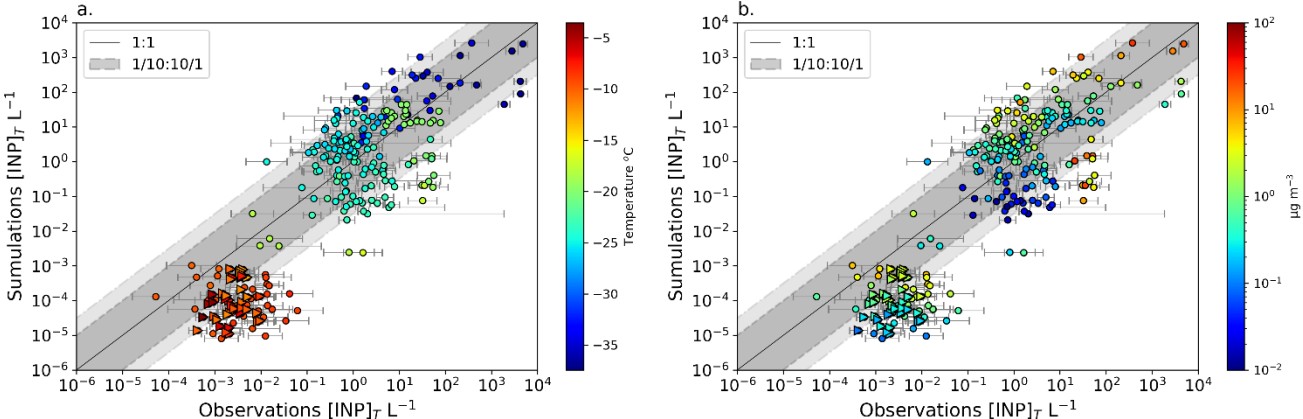

**Figure 4: Comparison of [INP]T concentrations calculated at the temperature of the measurements against observations (dates and locations provided in the supplementary Table S1). For each observation, the INP concentration is calculated at the temperature corresponding to the instrument temperature of the measurement. Circle markers correspond to the BACCHUS database and triangles to Wex et al. (2019) database. The dark grey dashed lines represent one order of magnitude of difference between modelled and observed, and the light-grey dashed lines 1.5 orders of magnitude. The simulated values correspond to monthly mean**
**concentration, and the error bars correspond to the observed error of monthly mean INP values. The color bars show the corresponding instrument temperature of the measurement in Celsius (a) and the modelled dust aerosol mass concentration in μg m-3 (b).**



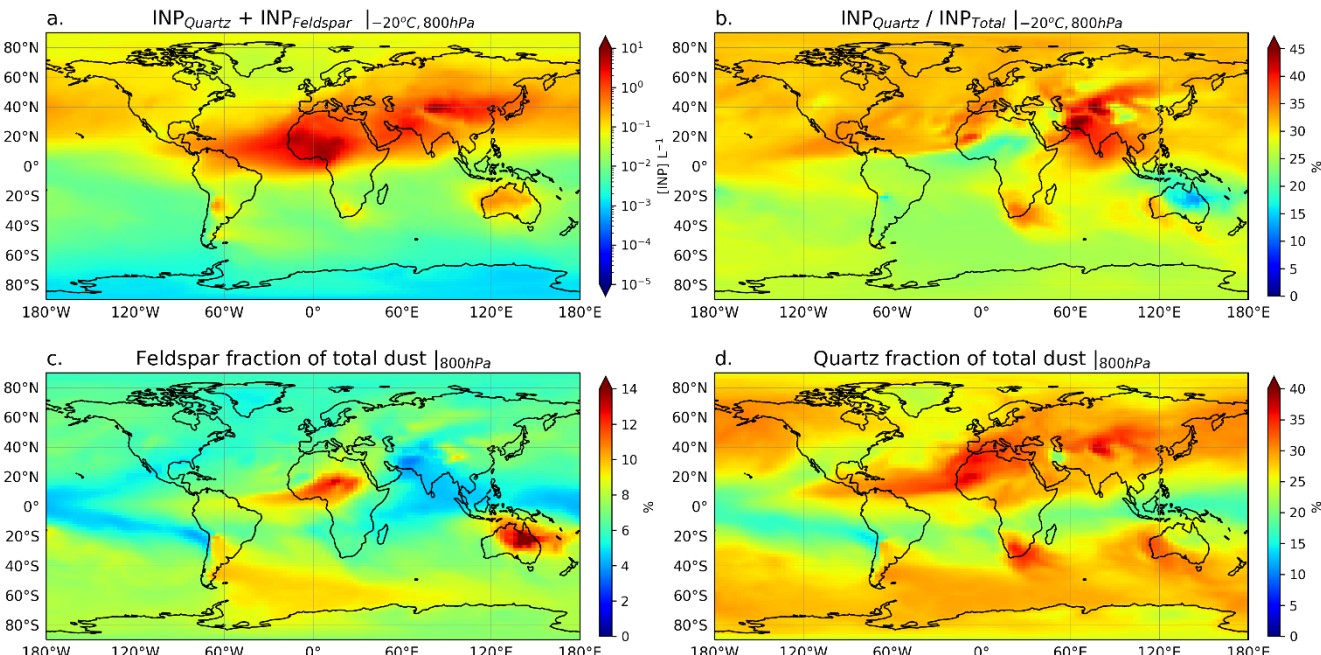

**Figure 5: Annual mean distributions calculated by TM4-ECPL for the year 2015 of (a) INP concentrations for an activation temperature of −20oC based on K-feldspar and quartz at a pressure level of 800 hPa, (b) ratio of INP from quartz over total INPs from dust (K-feldspar and quartz) at the same conditions, (c and d) mass ratios of feldspar (c) and quartz (d) in percentages over total dust mass at 800hPa.**

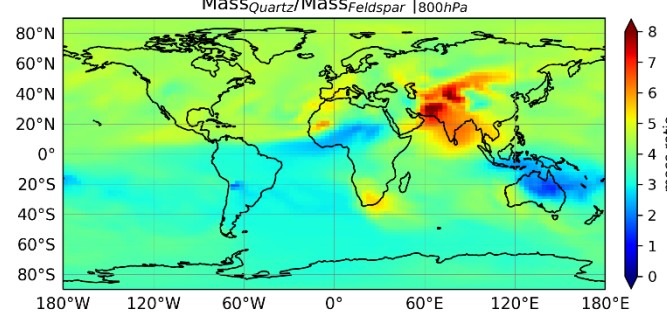

**Figure 6:  Ratio of quartz to feldspar minerals masses in dust as calculated by dividing quartz mass by feldspar mass.**





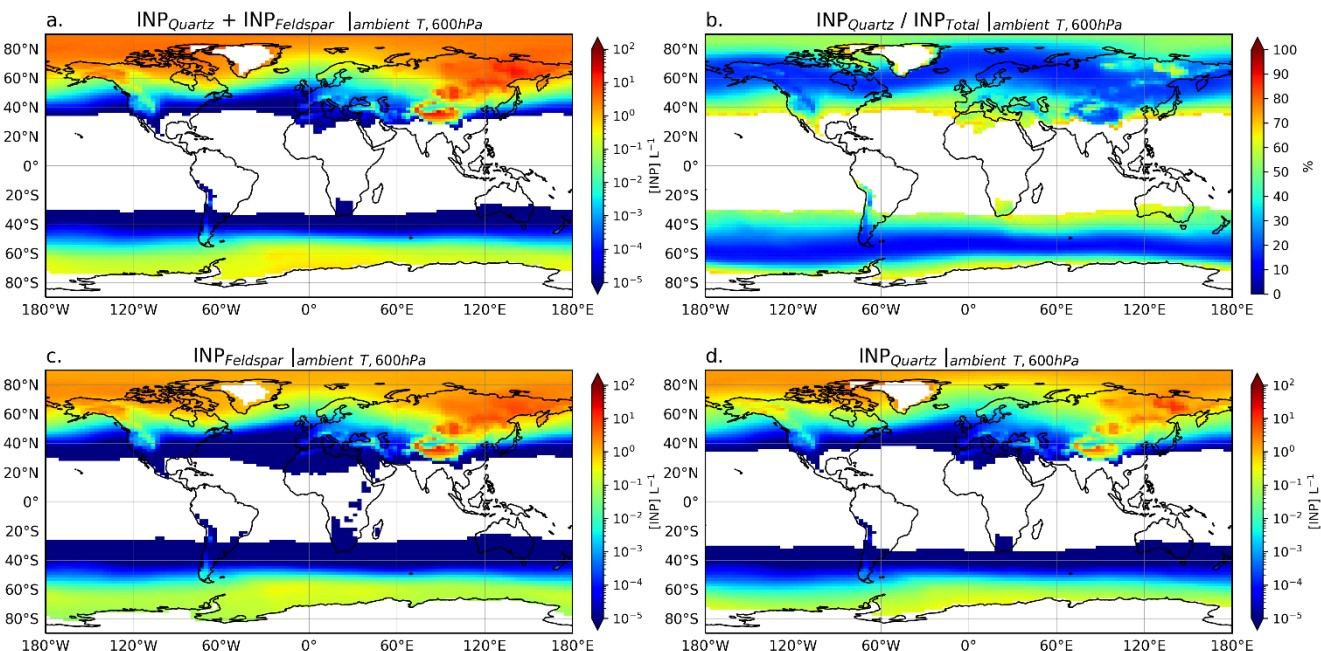

**Figure 7: Annual mean distributions of ice-nucleating particles number concentrations [INP]ambient calculated by TM4-ECPL at ambient (model's) temperature and 600 hPa based on the presence both of K-feldspar and quartz (a). Ratio of INP from quartz over total INP from K-feldspar and quartz at the same conditions (b). Panels (c-d) show the corresponding INP concentrations derived individually from each mineral: K-feldspar (c) and quartz (d).**

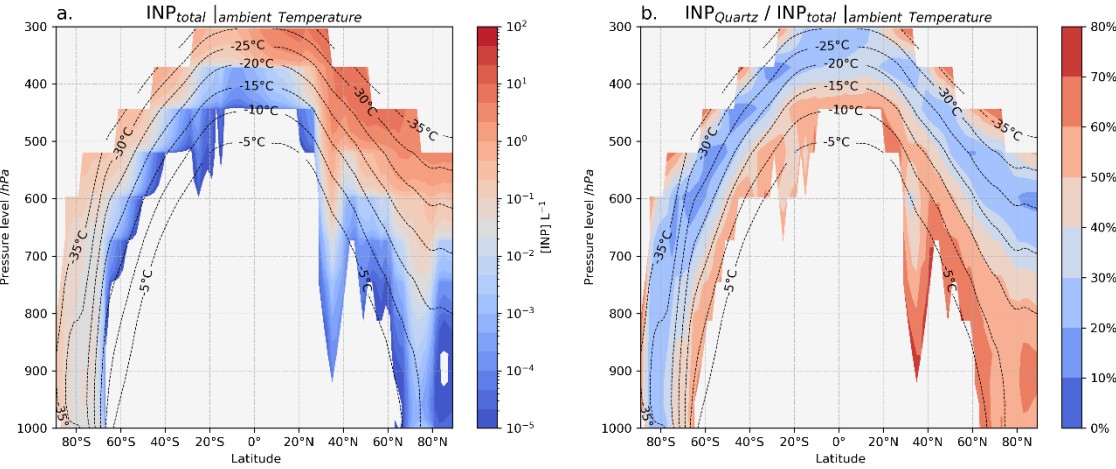

**Figure 8: Annual zonal mean profiles of [INP]ambient number concentration for 2015, calculated by TM4-ECPL and accounting for K-feldspar and quartz. The black contour dashed lines show the annual mean temperature of the model. The color map shows the INP concentration derived from feldspar and quartz dust aerosols [INP]total (a) and the ratio of INP from quartz to the total INP from both minerals (b).**