# Peer review of "Role of K-feldspar and quartz in global ice nucleation by mineral dust in mixed-phase clouds"

_Atmospheric Chemistry and Physics, 2022_

## Author Comment (AC1)

**Replies to reviewer #1**

We thank the reviewer for the careful reading of our manuscript and the positive comments. In the revised version, we have addressed all the comments of the reviewer, as detailed. below:

The reviewer's comments are provided in black and our replies in blue.

General comment:

The present study evaluated the importance of K-feldspar and quartz in ice nucleation and mixed-phase cloud formation. To do so, the authors used the global 3-dimensional chemistry-transport model TM4-ECPL where different parametrizations were added to fulfill the proposed objectives. The performance of the model was found to be satisfactory taking into account the associated uncertainties. The author found that besides K-feldspar, quartz can be a good source of INPs at specific locations, altitudes and temperatures with a good potential to impact mixed-phase and cirrus clouds. The present study provides new evidence on the importance of mineral dust particles in cloud formation. Additionally, the used approach can open future studies in the cloud physics community. Although the Introduction and Methodology sections are well written, this is not the case for Results and Discussion section. Therefore, the present manuscript can be accepted after the following comments are properly addressed.

Major comments:

There are several parts where the same information is repeated. I invite the authors to avoid redundancy.

We have revised the manuscript to avoid redundancies. For this, we have removed lines 184-186 and 196-198 and rephrased parts of sections 2, 3 and 4.

Figures are called in a random way making very difficult to follow the manuscript. Although most of the text is well written, the way Figures are incorporated into the text is not appropriate and needs to be fixed to improve de readability of the manuscript.

We have reordered the Figures to cite them in succession and relevant connecting text to improve the manuscript readability. Moreover, we refer to figures using parenthesis after the relevant discussion.

Section 4 is very confusing and needs to be deeply edited.

Section 4 has been restructured and heavily edited following the reviewer's. The text is now shorter with better flow. It reads as follows:

[revised manuscript text omitted]

Minor comments:

Lines 24-26: "Mineral dust emitted from arid regions, particularly potassium-containing feldspar (K-feldspar), has been shown to be the most efficient INP through immersion freezing in mixed-phase clouds." This is not completely true, as the most efficient INPs, in terms of the freezing temperature, are biological particles.

Good point. "the most efficient" has been replaced by ""a very effective." Following also the second reviewer's suggestion.

Line 34 and along the manuscript: low-level clouds)". With low-level clouds the authors mean liquid clouds? If yes, this sentence is not correct as the CCN abilities were not evaluated in the present study.

The whole study refers to mixed-phase clouds, where INP can have an impact on. Nevertheless, "low level clouds" has been modified to "low-level mixed-phase clouds" (that said, mixed-phase clouds do include droplets so CCN effects are also relevant there – even if not the topic of study here).".

Lines 48-49: "Ice nucleation in clouds proceeds by homogeneous freezing of liquid droplets at temperatures lower than −37°C" How about RHi?

Lines 49-50: "triggered by INP at warmer temperatures" How about RHi?

In order to refer also to RH conditions, the sentence has been modified as follows:

'Ice formation in clouds proceeds for supersaturation relative humidity with respect to ice (RHi) by both homogeneous and heterogeneous freezing at temperatures lower than −38°C (Liu et al., 2012) and by only heterogeneous freezing at warmer temperatures (DeMott et al., 2010; Murray et al., 2012; O'Sullivan et al., 2016; Price et al., 2018)

Line 64: "(e.g., Georgakaki et al., 2022; Sotiropoulou et al., 2021)". I suggest to cite older/pioneering studies.

Older references are now added (Field et al., 2016 provides a comprehensive review):

"Secondary processes (ice-ice collisions, droplet riming and droplet shattering) can modulate further these phenomena (e.g., Field et al., 2016; Georgakaki et al., 2022; Sotiropoulou et al., 2021; Field et

al., 2016) so that ice production is promoted even at extremely low INP concentrations (e.g., Morales Betancourt et al., 2012; Sullivan et al., 2018)."

Line 78: "have been proposed to act as INP". The word "proposed" is incorrect as there is very clear evidence that they can act as INP.

Good point. Changed to "have been identified to act as INP".

Line 108: I suggest to add a brief description about the immersion freezing definition.

A brief description has been added in this sentence: " Immersion freezing occurs when ice nucleation is initiated by an INP that becomes immersed in an aqueous solution or water droplet via activation of CCN during liquid cloud formation and it is thought to be the most dominant freezing mechanism for mixed-phase clouds (Kanji et al., 2017)."

Lines 151-153: How about the particle' density?

We added in this sentence: "and dust particle density is equal to 2650 kg m$^{-3}$".

Lines 178-180 and along the manuscript: Either use "parametrization" or "parametrisation"

We now use "parametrization" throughout the text.

Line 231 and along the manuscript: Figures need to be called chronologically, starting with Figure S1.

Figures have been reordered following the appearance in the manuscript.

Lines 310-311: 4b). As shown in Figure 4a, these points correspond to measurement temperatures around -20oC and -25oC. I do not get this.

The sentence has been rephrased for clarity: "As shown in Figure 4a, these low-dust mass points correspond to measurements performed at temperatures around -20$^o$C and -25$^o$C".

Line 351 and Figure 1: to activate and form INP. An aerosol particles has the capability or not to act as INP but INPs do not form.

Following the reviewer's suggestion, "form" has been replaced by "act as".

Line 368: I suggest to replace "there are quartz INP sources" to "could be quartz INP sources"

This part of the discussion has been rephrased as follows: "In contrast, the South Atlantic is affected by K-feldspar INP originating from the Patagonia desert since 75% of ice crystals potentially formed at -20$^o$C are derived from K-feldspar, and the remaining 25% is from quartz (Figure 5b). Additionally, high percentages of quartz minerals and low feldspar (about 6%) in terms of mass are calculated over South Africa originating from the Kalahari Desert. There 35% of the total INP concentrations (around 10$^{-2}$L$^{-1}$) originate from quartz particles (Figures 5c-d)."

Lines 373-379: I suggest to merge these lines with lines 362-365.

Reordering with significant improvement of the discussion of section 4 has been made, as seen in our replies above.

Line 383: 600hPa Why not at 800 hPa as in previous sections?

We have changed all figures to present the INP distributions at 600hPa uniformly.

Lines 388-389: "Total INP ([INP]total) maximizes approximately at 600-500hPa (Figure 8a)". I do not get this. At what latitude and temperature do the authors refer to?

This sentence has been rephrased as follows:

"Figure 8a shows the annual zonal mean distribution of primary ice crystals derived from both minerals. Total primary ice crystals ([INP]total) maximize approximately at 600-500 hPa below -25 °C in mid-latitudes at local model temperature (Figure 8a), with K-feldspar-derived ice crystals accounting for the largest fraction (Figure 8b). This behavior is attributed to the high ice activity of K-feldspar and its temperature dependence. However, at lower altitudes and below -12°C, low [INP]total concentrations ($<10^{-2}L^{-1}$) are calculated and are mainly derived from quartz dust particles (60%) (Figure 8b)."

Line 389: "where K-feldspar derived INP is the largest fraction of INP concentration". Again, at what latitude and temperature do the authors refer to?

See our reply to the previous comment of the reviewer.

In addition, we have drawn in Figure 1, here below, the percentage contribution of quartz-originating INP to the total INPs (from quartz and K-feldspar) at 34°N latitude and for all longitudes. We can see that at pressure levels between about 700 and 800 hPa and at around 80°E the contribution of quartz to INP concentration is about 60%. This is probably due to the large concentration of quartz particles compared to those of K-feldspar. Thus, even though at these temperatures (-10 °C to -15 °C), the active site density of quartz is very low, the number concentration of quartz particles is high, enhancing the concentration of INP originating from quartz. At higher altitudes in the atmosphere (i.e. lower pressure levels), the contribution of quartz to INP concentration decreases since K-feldspar is more active at this temperature range, and more K-feldspar particles are activated to form ice crystals. Finally, we can notice that at even higher altitudes in the atmosphere (e.g. 400-450 hPa), quartz becomes again an important contributor to INP concentration due to the significant particle concentration and its high active site density at the low temperatures of this pressure level (Harrison et al., 2019).

[Figure]

*Figure 1 : Annual mean of the percent contribution of INP from quartz minerals over 34 °N latitude across all longitudes.*

Lines 391-392: "are calculated at lower altitudes". Please indicate the temperature range.

See our reply to comments on lines 388-389.

Line 391: "quartz's high number concentration". Please cite literature indicating that the Gobi Desert is a good source of quartz.

We have rephrased this part of the discussion. The sentence now reads:' "This is attributed to quartz's high number concentration between 30-40oN at and below 700 hPa, partially associated with higher local quartz than feldspar emissions from Asian dust sources (Claquin et al.,1999; Nickovic et al., 2012)'".

Lines 394-395: At temperatures below −25°C, the quartz contribution becomes increasingly important with decreases in temperature at high INP concentrations. I do not get this.

This sentence was rephrased: "At temperatures below −25°C, the quartz contribution becomes increasingly important with increases in INP concentration when the temperature decreases, reaching up to 50% at −35°C (Figure 8b)."

Lines 400-401: and is far from emission sources. It is unclear to me.

This sentence has been rephrased.

Line 401: "Figure S5 for Eurasia". This is the only mention for this Figure and it deserves to be deeper discussed

Figure S5 is now Figure S6. Some discussion is now provided. We also discuss Figure S7: "This is clearly demonstrated in Supplementary Figure S6 for Eurasia where at the range of 700-900hPa and around 450 hPa model pressure levels the contribution of [INP]$_{quartz}$ to total ice crystals from immersion freezing on dust particles exceeds 60%. Figure S7 shows that in the South Hemisphere quartz contribution to total INP is about 40%. Consequently, [INP]$_{feldspar}$ is expected to affect mid-altitude clouds, while [INP]$_{quartz}$ is expected to affect both the low-altitude clouds and the high-altitude cold clouds"

Lines 423-424: "Consideration of quartz-derived INP is improving the comparison with observations for high INP concentrations at low temperatures and relatively low INP concentrations at temperatures around -20oC". I do not get this.

In section 3 it is written: "Figure S4d clearly shows that consideration of quartz together with Feldspar INP improves the comparison with observations by moving the respective points closer to the 1:1 line. This is particularly true for high INP concentrations at low temperatures (below -30ºC) and for relatively low INP concentrations at temperatures around -20ºC".

The sentence to which the reviewer refers in the Conclusion section. It now reads as follows:

"Consideration of quartz-derived INP improves the comparison with observations for high INP concentrations at low temperatures (below about -30ºC) and relatively low INP concentrations at temperatures around -20ºC."

Line 434: The H-M mechanism usually takes place around -5°C, therefore, it may not be of high importance for the temperatures evaluated here.

This sentence was rephrased: "As importantly, the heterogeneously formed ice crystals could trigger secondary ice production processes, such as the Hallett – Mossop process, which usually take place around -5°C and could improve our comparison at high temperatures. These processes are not considered in our study. However, they are known to further increase the concentration of ice crystals (Sotiropoulou et al., 2020) and their subsequent climate impact."

Figure 3: From the figure caption my understanding is that panel (a) should have two colors only.

Figure 3 caption has been improved by adding the following explanation: "Colors indicate measurement locations listed in the figure legend."

Figure 4: "[INP]T" and "m-3". Fix this.

Done.

Figure 6 and along the manuscript. Are the authors referring to K-feldspar or all types of feldspar?

All types of feldspar.

Figure S3: The Figure legend covers some sampling sites

This figure (new Figure S1) was redrawn to move the legend next to the figure.

Figure S7: Should "This figure is related to Fig 3." Be "This figure is related to Fig 4."?

Figure S7 is a new Figure S4 and is related to Figure 4, as mentioned in the revised caption.

Table S3: This is not mentioned in the main text

Table S3 is now mentioned in the caption of Figure S1: The name of the campaigns for these datasets, the locations and the corresponding literature references are provided in Table S3.

Technical comments:

Line 17: It should be Switzerland
Corrected.

Line 57: Add a reference after "forcing"

(Vergara-Temprado et al., 2018) has been added.

Line 82: I suggest to replace Hoose et al. (2010a) by Hoose and Mohler (2012).

Replaced.

Line 92: Add a reference after "density"

Harrison et al. (2019) has been added.

Line 109: Westbrook and Illingworth, 2013 I suggest to add a more appropriate reference here, and additionally, it is out of place.

The sentence now reads: "Immersion freezing occurs when ice nucleation is initiated by an INP that becomes immersed in an aqueous solution or water droplet via activation of CCN during liquid cloud formation, and it is thought to be the most dominant freezing mechanism for mixed-phase clouds (Kanji et al., 2017).'".

Line 124: Add a reference after "scavenging"

Abdelkader et al. (2017) has been added.

Line 166: Add a reference after "sieving"

Boose et al. (2016) has been added.

Line 180: Replace "as ice nuclei particles" with "INP"

Replaced.

Line 190: Add a reference after "surfaces"

When condensing text, this sentence has been removed.

Line 245: Replace "3.2" with "Section 3.2"

Replaced.

Line 278: "2022) _databases" Fix this.

Done.

Line 314: "low temperatures". Specify low temperatures.

We direct the reader to section 2.3, where the temperature limits of the parametrizations are provided.

Line 337: "see also Supplementary Figure S8)" There is not Figure S8 in the supplementary material.

We now refer to supplementary Table S2.

Line 379: "°C". Fix this.

Done.

Line 396: Should "(Figure 8a)" be "(Figure 8b)"?

Corrected.

Lines 399, 416, 417: Add a reference after "K-feldspar"

This part of the discussion has been rephrased.

Line 414: "k-feldspar". Fix this.

Done.

Line 419: What do the authors mean with "global atmosphere"?

"Global" has been removed.

---

## Author Comment (AC2)

**Replies to reviewer #2**

We thank the reviewer for the careful reading of our manuscript and the positive comments. In the revised version, we have addressed all the comments of the reviewer, as detailed. below:

**The reviewer's comments are provided in black and our replies in blue.**

This study by Chatziparaschos et al. used a 3D chemical transport model to predict global concentrations of ice nucleating particles based on an emission model accounting for quantities of quartz and feldspar emitted and parameterisations of the ice nucleating effectiveness of quartz and feldspar. The results of this process are shown to agree reasonably well with field measurements of ice nucleating particle concentrations. The study concludes that the high abundance of quartz particles means that their contribution to INP concentrations is significant, despite the lower ice nucleation ability of quartz compared to 'K-feldspar'. Conclusions are also drawn about the contributions of the different minerals to low and mid-level clouds. The study is interesting and appears to be well-conducted. I have a few suggestions below but support publication once these are considered. While mostly well-written, the study would benefit from thorough proofreading. I have highlighted various issues below but there are likely more.

**General comments:**

I think it is worth noting that a great deal is still not known about why quartz and feldspar nucleate ice. In the absence of proper physico-chemical understanding of how these minerals nucleate ice the parameterisations of Harrison et al. remain entirely empirical. It may still turn out that they are not representative for unforeseen reasons. This doesn't detract from the study but is probably worth mentioning briefly. Relatedly, I think a few words clarifying why it is alkali feldspar (I think use of K-feldspar throughout is fine, although worth noting that some Na-rich feldspar also nucleate ice well), rather than plagioclase feldspars, that are important may be helpful e.g (Kiselev et al., 2021;Whale et al., 2017;Harrison et al., 2016).

Discussion has been added, and the reference by Harrison et al. (2016) has been introduced:

"More recently, it was shown that among dust minerals, alkali feldspars, in particular the potassium feldspars, nucleate ice more efficiently than feldspars in the plagioclase series, which contain calcium (Harrison et al., 2016). Also, sodium-rich feldspars are very ice-active but were found to lose their ice-activity with time compared to K-feldspars (Harrison et al., 2016), thus, K-feldspar is the most efficient INP. Note, however, that little is known about the processes governing the IN capability of these minerals, so more future work is needed to unravel their effect and contribution for all possible ice formation conditions (Burrows et al., 2022)".

The abbreviation INP, is used in various ways throughout. While what is meant is mostly clear it would probably be best to pick a definition and stick with it.

We now use INP for particles having the potential to activate and form ice at a specific temperature, and once activated, we talk about ice crystals.

Suggest checking the format of references throughout the text, there is some variation.

**We have checked and corrected the format of the references.**

**Specific comments**

Line 23 – INP usually abbreviates Ice Nucleating Particle, with an 's' added for 'Ice Nucleating Particles'

**Done.**

Line 23 is confusing. Ice nucleation is the first step of ice formation, remove reference to 'homogeneous formation'

This sentence has been rephrased as follows: "Ice formation is enabled by Ice Nucleating Particles (INPs) and can profoundly affect the microphysical and radiative properties, lifetimes, and precipitation rates of clouds."

Lines 25 – The word 'efficient' implies a ratio. 'Effective' or similar would be better here.

**"Effective" is now used.**

Line 26 – It isn't clear what is meant by 'ice nuclei' (Vali et al., 2015). 'Ice nucleation activity' would be more typical I think.

**Done.**

Line 34 – I wonder if it might be possible to briefly explain why differences are seen between cloud regimes? This isn't really discussed in the text either. What causes the differences between quartz and k-feldspar as regards the cloud types they influence?

**The relevant sentence in the abstract has been modified as follows:**

"Our results show that although K-feldspar remains the most important contributor to INP concentrations globally, affecting mid-level clouds, the contribution of quartz can also be significant. It dominates at the lowest and the highest altitudes of dust-derived INP, affecting mainly low-level and high-level mixed-phase clouds."

And the relevant discussion in section 4: "However, at lower altitudes and below -12°C, low [INP]total concentrations (<10-2L-1) are calculated and are mainly derived from quartz dust particles (60%) (Figure 8b). This outcome is attributed to quartz's high number concentration between 30-40°N at and below 700 hPa, partially associated with higher local quartz than feldspar emissions from Asian dust sources (Claquin et al.,1999; Nickovic et al., 2012). At temperatures below –25 °C, the quartz contribution becomes increasingly important with increases in INP concentration when the temperature decreases, reaching up to 50 % at –35 °C (Figure 8b). These findings agree well with Ilić et al. (2022) and Boose et al. (2016), who showed that quartz could significantly contribute to [INP] at temperatures between the homogeneous freezing limit and –33°C. Overall, [INP]quartz dominates at the lowest and the highest altitudes of dust-derived INP (Figure 8b; reddish colors). This is clearly demonstrated in Supplementary Figure S6 for Eurasia where at the range of 700-900hPa and around 450 hPa model pressure levels the contribution of [INP]quartz to total ice crystals from immersion freezing on dust particles exceeds 60%. Figure S7 shows that in the South Hemisphere quartz

contribution to total INP is about 40%. Consequently, [INP]feldspar is expected to affect mid-altitude clouds, while [INP]quartz is expected to affect both the low-altitude clouds and the high-altitude cold clouds..'

See also the discussion of Figure 1 provided above, that explains how the combined effect of mineral concentration and its active sites at different pressure and temperature levels changes with height.

Line 71 – '....concentration of INP' maybe?

Modified as suggested.

Line 86 – Doesn't read well if INP means 'Ice nucleating particles'

This sentence was rephrased to address the comments of the other reviewer.

Line 89-90 - Kiselev et al. was in 2017 I think?

**The reference was corrected.**

Line 105 – The singular approximation assumes that each droplet contains a single ice nucleating particle active in a given temperature interval. I don't think the statement regarding site density is necessarily true.

This sentence was rephrased to include the reviewer's comment: "This approach adopts the socalled singular hypothesis that assumes that each droplet contains a single ice nucleating particle active in a given temperature interval and the time dependence (stochasticity) of nucleation is of secondary importance (Vali et al., 2015)."

Line 125 – may be worth briefly noting that solution environment may well substantially impact the ice nucleation effectiveness of both quartz and feldspar e.g. (Kumar et al., 2019;Whale et al., 2018;Klumpp et al., 2022;Whale, 2022).

We added further explanation following the above suggestion: "In addition, the solution environment may substantially impact the ice nucleation efficiency of both quartz and feldspar (Whale et al., 2018; Kumar et al., 2019), but the study of the impact of this process on INP is beyond the scope of the present study."

Line 187 – I wouldn't call the samples used 'soil'. Mostly they are mineral samples that have been selected for purity. The point that the samples may not be representative of atmospheric conditions is very true however.

We have replaced "soil" with "mineral" as suggested by the reviewer.

Line 339 – I don't think Spracklen and Heald looked at marine organic aerosol?

Spracklen and Heald's ref was misplaced. The sentence now reads: "These differences between model results and observations could be attributed to several factors, such as the omission of the contribution of marine organic (Wilson et al., 2015) and terrestrial biogenic aerosols (Spracklen and Heald, 2014; Myriokefalitakis et al., 2021)."